# Identification of Existing Challenges and Future Trends for the Utilization of Ammonia-Water Absorption–Compression Heat Pumps at High Temperature Operation

Marcel Ulrich Ahrens [1,*], Maximilian Loth [2], Ignat Tolstorebrov [1], Armin Hafner [1], Stephan Kabelac [2], Ruzhu Wang [3] and Trygve Magne Eikevik [1]

[1] Department of Energy and Process Engineering, Norwegian University of Science and Technology, 7491 Trondheim, Norway; ignat.tolstorebrov@ntnu.no (I.T.); armin.hafner@ntnu.no (A.H.); Trygve.m.eikevik@ntnu.no (T.M.E.)

[2] Institute of Thermodynamics, Leibniz University Hannover, Am Welfengarten 1, 30167 Hannover, Germany; loth@ift.uni-hannover.de (M.L.); kabelac@ift.uni-hannover.de (S.K.)

[3] Institute of Refrigeration and Cryogenics, Shanghai Jiao Tong University, Shanghai 200240, China; rzwang@sjtu.edu.cn

* Correspondence: marcel.u.ahrens@ntnu.no; Tel.: +49-1729237476

**Abstract:** Decarbonization of the industrial sector is one of the most important keys to reducing global warming. Energy demands and associated emissions in the industrial sector are continuously increasing. The utilization of high temperature heat pumps (HTHPs) operating with natural fluids presents an environmentally friendly solution with great potential to increase energy efficiency and reduce emissions in industrial processes. Ammonia-water absorption–compression heat pumps (ACHPs) combine the technologies of an absorption and vapor compression heat pump using a zeotropic mixture of ammonia and water as working fluid. The given characteristics, such as the ability to achieve high sink temperatures with comparably large temperature lifts and high coefficient of performance (COP) make the ACHP interesting for utilization in various industrial high temperature applications. This work reviews the state of technology and identifies existing challenges based on conducted experimental investigations. In this context, 23 references with capacities ranging from 1.4 kW to 4500 kW are evaluated, achieving sink outlet temperatures from 45 °C to 115 °C and COPs from 1.4 to 11.3. Existing challenges are identified for the compressor concerning discharge temperature and lubrication, for the absorber and desorber design for operation and liquid–vapor mixing and distribution and the choice of solution pump. Recent developments and promising solutions are then highlighted and presented in a comprehensive overview. Finally, future trends for further studies are discussed. The purpose of this study is to serve as a starting point for further research by connecting theoretical approaches, possible solutions and experimental results as a resource for further developments of ammonia-water ACHP systems at high temperature operation.

**Keywords:** industrial heat pump; high temperature heat pump; absorption–compression heat pump; natural refrigerant; ammonia-water; solution circuit

## 1. Introduction

Decarbonization of the industrial sector is one of the most important keys to reducing global warming. Energy demands and associated greenhouse gas (GHG) emissions in various industrial processes are continuously increasing [1]. Simultaneously, large amounts of potentially usable waste heat are available [2–4]. With climate change being one of the most significant topics of modern society, it is now globally recognized that there is a need to increase the energy efficiency of industrial processes and reduce direct GHG emissions, e.g., from burning of fossil fuels, in order to achieve environmentally friendly, cheap and sustainable energy systems [5–7]. Many industries requiring both cooling and heating

currently have separate systems for these tasks. Having a combined system capable of providing for both demands would be much more energy efficient. Due to this situation, the integration of high temperature heat pumps (HTHPs) with natural refrigerants is a promising approach for many industrial applications [8,9].

Following the trend towards more efficient and environmentally friendly ways of providing thermal energy from available waste heat as usable heat for industrial applications, suitable HTHP solutions have been increasingly investigated in recent years [10,11]. Many advances have been accomplished within the heat pump technology, in particular at lower heat delivery temperatures but a few at high temperatures above 90 °C [12,13]. However, ongoing research is seeking to further increase the delivery temperatures, as great demand and utilization potential exists for high temperature applications up to 150 °C due to the prevailing industry conditions [14]. Here, the use of natural refrigerants with low global warming potential (GWP) and known impacts and burdens on the atmosphere, such as ammonia and water, is of particular interest regarding environmental sustainability [15–17]. Therefore, the ammonia-water absorption–compression heat pump (ACHP) is considered a promising approach for heat pump applications in high temperature operations and is presented in more detail below.

The ACHP is often named a vapor compression cycle with solution circuit, compression/absorption cycle or hybrid absorption–compression heat pump. It combines the technologies of an absorption and vapor compression heat pump using a zeotropic mixture of ammonia and water as working fluid. As a result, heat is extracted and released at non-constant temperature glides. The required compression ratio can be reduced, when compared with conventional vapor compression heat pumps (VCHPs) utilizing single fluid refrigerants, by adjusting the concentration to the given boundary conditions. These characteristics, combined with the ability to achieve high sink temperatures above 100 °C at large temperature lifts and high coefficients of performance (COP), make the ACHP system a valuable solution for high temperature heat supply in the industry [18].

In 1895, the first patent concerning ACHP cycles was published by Osenbrück [19]. Detailed theoretical studies were first conducted by Altenkirch in 1950 and indicated a significant potential for energy savings [20]. Due to the energy saving potential and the increasing urgency to substitute the ozone-depleting chlorofluorocarbons (CFCs) combined with the energy crisis in the 1970s, research activities have increased rapidly since the 1980s, and several experimental plants have been built in this context. In 1997, Groll [21] summarized the research activities by reviewing more than 40 papers in a detailed overview. It was concluded that, despite the investigation of various cycle configurations and the commissioning and operation of several large-scale pilot plants, considerable work remained to be done before the ACHP could be used commercially.

In the following years, research activities continued and were stimulated by the increasing energy demand in the industrial sector with growing awareness of GHG emissions and the increased motivation for HTHP solutions using natural refrigerants. During the installation of further test and pilot plants, the ACHP with a single-stage solution circuit was successfully brought to commercial implementation using standard refrigeration components [22,23]. Until today, several units have been installed for commercial use in various industrial applications, achieving heat sink temperatures of up to 120 °C and temperature lifts of up to 75 K [24,25].

The first commercial installations of the ACHP encouraged the growing interest in the system. They have led to extensive theoretical investigations in recent years to identify optimal operating conditions and potential applications [26–29]. Furthermore, particular focus was placed on possible improvements to further expand the achievable process parameters, such as the sink outlet temperature and system efficiency, to compete with conventional solutions for use in high temperature applications [30–32].

The present work aims to support the current trends in the scientific field, focusing on developing and improving capable HTHP solutions for the use in industrial high temperature applications. For this reason, this study identifies the existing challenges and future trends for the utilization of the ACHP at high temperature operation against the background of recent research activities and findings. First, the ACHP cycle and various modifications are presented. This is followed by a comprehensive review of experimental work related to the described cycle modifications and the identification of existing challenges. Then, current developments and possible solutions are described and presented in a detailed overview. Finally, the future trends of research and innovation activities are defined based on the performed investigations.

## 2. The Ammonia-Water Absorption–Compression Heat Pump

The most basic type of the ACHP cycle using ammonia-water as a working fluid is the Osenbrück cycle, named after its inventor [19]. ACHP cycle with a single-stage solution circuit consists of seven main components: Three heat exchangers, a liquid–vapor separator, an expansion valve, a solution pump and a compressor. Figure 1 shows a simplified representation of this cycle.

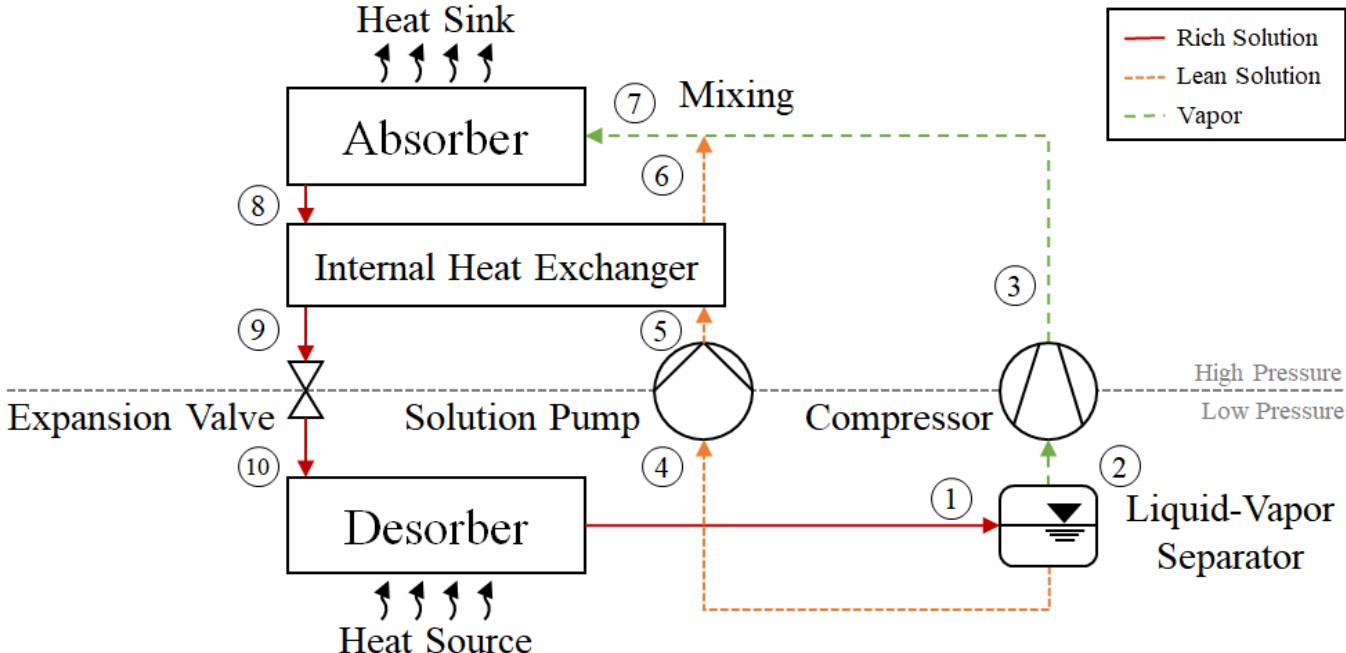

**Figure 1.** Schematic representation of a basic hybrid absorption–compression heat pump cycle.

In the ACHP cycle, the evaporator and condenser are replaced by desorber and absorber. Throughout the entire heat transfer process with the heat source in the desorber, the ammonia-water mixture is in the two-phase region and the ammonia concentration in the liquid phase decreases. Due to incomplete evaporation, a two-phase mixture leaves the desorber (1). A liquid–vapor separator is applied to separate the phases and ensure that only vapor enters the compressor (2) to be compressed to the high-pressure side of the cycle (3). The liquid phase, characterized as lean solution, is sent to the solution pump (4) where the pressure level increases (5). Then, the lean solution passes through an internal heat exchanger (IHX) to increase liquid temperature and improve overall cycle performance (5,6). After the IHX and compressor, the liquid and vapor streams are mixed, resulting in a liquid–vapor mixture (7). In the absorber, vapor is absorbed into the liquid phase rejecting heat to the heat sink. The ammonia concentration in the liquid phase gradually increases so that the saturated liquid leaving the absorber at the outlet is called rich solution (7,8). Heat is transferred from the rich

to the lean solution stream in the IHX (8,9) before the solution is throttled to the low-pressure level (9,10) returning to the desorber and completing the cycle.

Ammonia-water mixture as working fluid with a large boiling point difference and the implementation of the solution circuit provides two additional degrees of freedom compared to conventional VCHP cycles [21]. The ability to vary the working fluid composition and the circulation ratio between solution pump and compressor streams ensures high flexibility and adaptability of the operating parameters by changing boundary conditions. In general, the following advantages for the ACHP system can be pointed:

- Capacity control by changing the overall composition of the working fluid mixture, resulting in a change in the low-pressure gas density. Hereby, at constant speed and volume flow of the compressor, the mass flow of the vapor and thus the capacity of the heat pump is changed.
- Exploitation of the occurring temperature glides in desorber and absorber can be matched to heat source and sink and thus reduce the irreversibility of the system and enable large temperature spans with comparatively high COPs. The process follows the Lorenz rather than the Carnot process and becomes more effective as the temperature spread increases.
- Compared to pure ammonia, higher heat sink temperatures can be achieved with lower discharge vapor pressure and reduced pressure ratios when water is used as solvent.

*Cycle Configurations of the Absorption–Compression Heat Pump*

Based on the ACHP cycle introduced in Figure 1, several authors such as Amrane et al. (1991) [33], Hultén and Berntsson (2002) [34] and Jensen (2015) [18] examined various process modifications. These included extra components, such as additional heat exchangers for the internal heat exchange or cooling circuits. As a result, the system could be further adapted and improved, but the complexity has also increased. In addition to these modifications of the ACHP cycle with single-stage solution circuit, other advanced cycle configurations have been developed. Figure 2 shows three cycle configurations of the ACHP, which have been assigned individual designations due to the modifications made and specific characteristics.

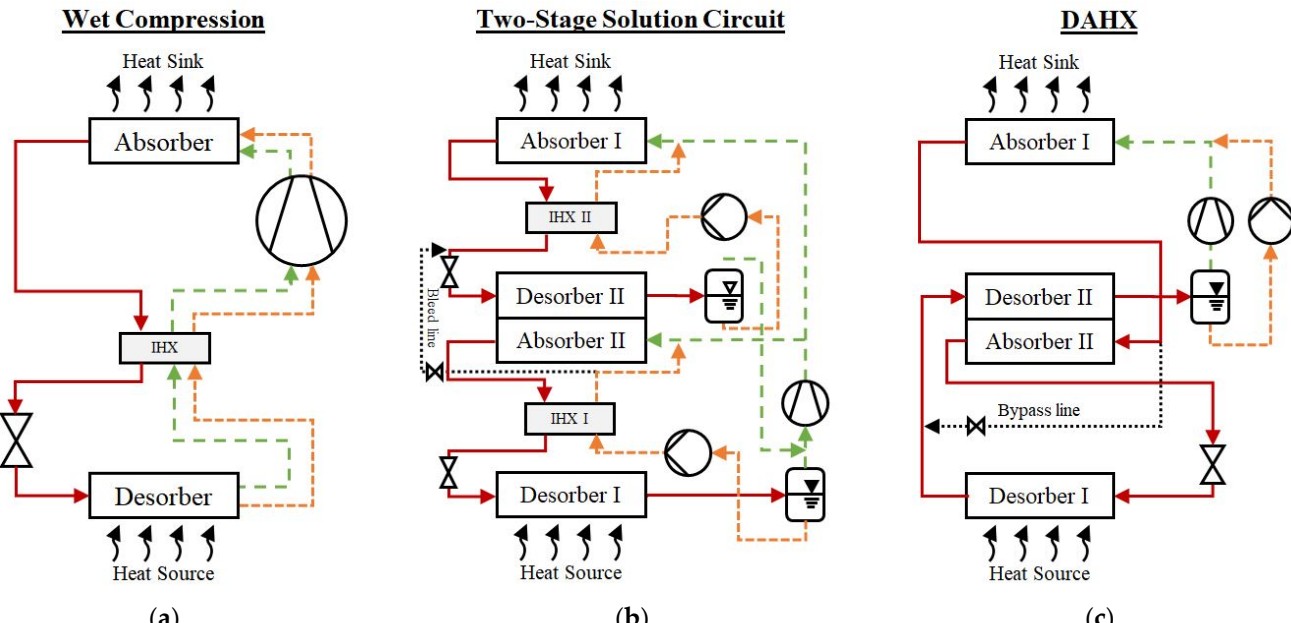

**Figure 2.** Schematic representations of (**a**) a wet compression cycle, (**b**) ACHP with two-stage solution circuit and (**c**) ACHP with single circuit and desorber/absorber heat exchange (DAHX).

The wet compression cycle, also known as wet compression-resorption cycle, is shown in Figure 2a and was investigated by Bergmann and Hivessy (1990) [35], Itard and Machielsen (1994) [36] and Itard (1995) [37]. Contrary to dry compression with a separate solution circuit, the ammonia-water mixture leaving the desorber in a wet-vapor state is conveyed to the compressor without additional phase separation. Thus, a two-phase compression takes place, intended to reduce the compressor discharge temperature and the required compression work by decreasing the superheating of the vapor phase through continuous cooling by the present liquid. To achieve a good system performance, an extensive internal heat exchange and a suitable compressor with high isentropic efficiency are important. A disadvantage of the basic wet compression approach is the loss of flexibility since the circulation ratio and mixture composition can no longer be varied during operation.

In order to keep the flexibility of the ACHP cycle combined with the advantages of increased internal heat exchange, the modification of the single-stage solution circuit towards the two-stage solution circuit, as shown in Figure 2b, was investigated by Radermacher (1988) [38], Rane and Radermacher (1991) [39] and (1993) [40]. Here, two solution circuits are staggered, connected with an intermediate absorber–desorber pair for internal heat exchange. Occurring concentration differences between the solution circuits are compensated by a bleed line. The lean solution from the low temperature circuit is fed into the rich solution of the high temperature circuit. This modification and the reduced temperature differences for each stage enable the system to achieve high temperature lifts of up to 100 K at comparatively modest pressure ratios, which can be achieved with a single compressor stage. Compared to a single fluid VCHP, a reduction in the required pressure ratio of 40% to 65% and a resulting improvement in COP can be achieved for a given temperature lift [41]. However, because both stages are supplied with vapor by the compressor, the capacity will be reduced by 45%. This leads to a necessary increase in mass flow, which however can have a positive effect on the selection and performance of a suitable compressor.

The vapor compression cycle with solution circuit and desorber/absorber heat exchange (DAHX) cycle, as shown in Figure 2c, was investigated by Groll and Radermacher (1994) [42]. In this approach, the gliding temperature intervals of desorber and absorber are further increased, allowing them to overlap. A portion of the heat transferred internally from absorber II to desorber II. As a result, the DAHX cycle only requires one solution pump and compressor while having similar characteristics to a two-stage solution circuit. Due to the internal heat exchange, the required pressure ratio for a temperature lift of 75 K can be reduced by up to 75% compared to a single fluid VCHP, with a possible COP increase of more than 40%. However, using only one solution circuit results in a loss of flexibility regarding the temperature glides, whereby the temperature intervals are dependent on the pressure ratio and can no longer be selected independently.

Figure 3 shows a simplified comparison of the described ACHP cycles for the operation in a defined temperature interval in a ln p–1/T diagram for ammonia-water mixture.

Here, all cycles start for identical composition and given source temperature at the same desorber pressure. The single-stage ACHP cycles require the largest pressure ratio to reach the desired sink temperature. The ACHP with two-stage solution circuit features two separate heat transfer arrows due to the extra absorber–desorber pair and reaches the absorber temperature at a lower pressure ratio. The DAHX cycle with the linked heat exchange requires the lowest pressure ratio.

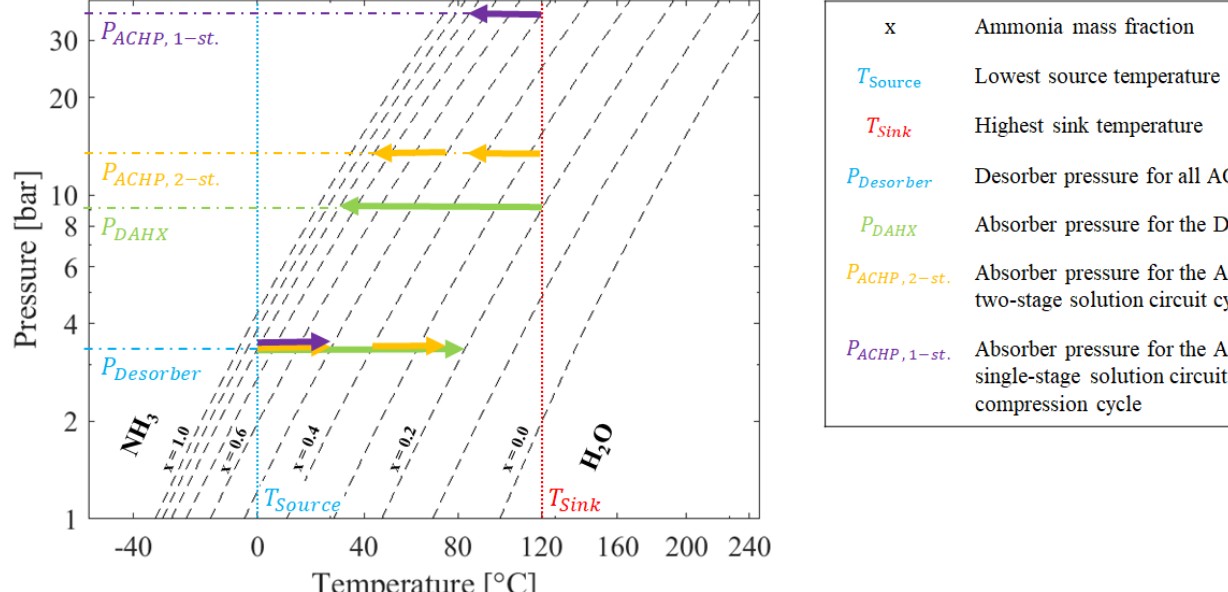

**Figure 3.** Simplified comparison of the described ACHP cycles in a ln p–1/T diagram (adapted from Groll (1997) [18]).

Based on the presentation of the cycle modifications, it can be concluded that each cycle provides specific advantages and associated application areas. The ACHP with single-stage solution circuit is considered most suitable for conventional applications with the advantage of adjustable temperature glides and capacity. Wet compression can help to solve occurring problems such as high compressor discharge temperatures. The two-stage solution circuit is considered a promising approach for applications with the highest temperature lift and small temperature glide, while the DAHX cycle achieves considerable temperature glides with lower temperature lift [43].

## 3. State of Technology

In the following, the state of technology for the ACHP system using ammonia-water mixture as working fluid will be examined based on the experimental work carried out. The experiments conducted until 1994 were comprehensively discussed by Groll (1997) [21] with an update of experiments until 2001 by Nordtvedt (2005) [44]. Based on these results, Table 1 presents an updated overview of conducted experimental investigations.

**Table 1.** Overview of experimental investigations on ammonia-water absorption–compression heat pump cycles.

| Author (Location) | Year | Cap. [kW] | $T_{source}$ [°C] | $T_{sink}$ [°C] | $\Delta T_{m,lift}$ [K] | COP [-] | Compressor Design | Absorber Design | Desorber Design | Ref. |
|---|---|---|---|---|---|---|---|---|---|---|
| colspan across | | | | | **ACHP with single-stage solution circuit** | | | | | |
| Bercescu et al. (Romania) | 1983 | 15 | 24 | 59 | n/a | 4.9 | oil, recip, t-st | n/a | n/a | [45] |
| Pop et al. (Romania) | 1983 | 4500 | 36 to 30 | 45 to 55 | 16.9 | 2.6 | dry, recip, s-st | Shell/tube horizontal | Shell/tube horizontal | [46,47] |
| Mučić, Scheuermann (Germany) | 1984 | 160 | 60 to 45 | 25 to 78 | −5.6 | 11.3 | oil, recip, s-st | Shell/tube horizontal | Shell/tube horizontal | [48] |
| Stokar, Trepp (Switzerland) | 1987 | 15 | 40 to 15 | 40 to 70 | 28.1 | 4.3 | dry, recip, s-st | Shell/tube vertical | Shell/tube vertical | [49] |
| Mučić (Germany) | 1989 | 1000 | 95 to 95 (const.) | 115 to 115 (const.) | 20.0 | 9.1 | dry, screw, s-st | Shell/tube vertical | Shell/tube vertical | [50] |
| Rane et al. (USA) | 1989 | 7 | 15 (avg.) | 45 (avg.) | n/a | 3.7 | dry, recip, t-st | Shell/tube vertical | Shell/tube vertical | [51] |
| Baksaas, Grandum (Norway) | 1999 | 60 | 48 | 48 to 100 | n/a | 2.1 | oil, recip, t-st | Corrugated PHE | Corrugated PHE | [22] |
| Mongey et al. (North Ireland) | 2001 | 13.5 | 42 to 27 | 42 to 57 | 15.2 | 3.7 | oil, recip, s-st | Corrugated PHE | Corrugated PHE | [52] |
| FKW (Germany) | 2003 | 27 | 43 to 35 | 60 to 72 | 27.0 | 4.3 | oil, twin screw, s-st | Corrugated PHE | Corrugated PHE | [53] |
| Risberg et al. (Norway) | 2004 | 300 | 50 to 15 | 50 to 85 | 36.9 | n/a | oil, recip, t-st | Corrugated PHE | Corrugated PHE | [23] |
| Nordtvedt (Norway) | 2005 | 47 | 50 to 17 | 50 to 93 | 38.7 | 2.4 | oil, recip, t-st | Corrugated PHE | Corrugated PHE | [44] |
| Nordtvedt et al. (Norway) | 2011 | 650 | 48 to 38 | 48 to 87 | 22.8 | 4.5 | oil, recip, t-st | Corrugated PHE | Corrugated PHE | [24] |
| Kim et al. (Republic of Korea) | 2013 | 10 | 50 to 30 | 50 to 90 | 28.9 | 3.0 | oil, recip, t-st | Corrugated PHE | Corrugated PHE | [54] |
| Jung et al. (Republic of Korea) | 2014 | 7.3 | 50 to 30 | 50 to 81 | 25.1 | 2.7 | oil, recip, s-st | Corrugated PHE | Corrugated PHE | [55] |
| Markmann et al. (Germany) | 2019 | 40 | 59 to 49 | 50 to 60 | 1.0 | 2.5 | oil, twin screw, s-st | Corrugated PHE | Corrugated PHE | [56] |
| Ahrens et al. (Norway) | 2021 | 940 | 67 to 60 | 73 to 95 | 20.1 | 5.9 | oil, recip, t-st | Corrugated PHE | Corrugated PHE | [25] |

Table 1. *Cont.*

| Author (Location) | Year | Cap. [kW] | $T_{source}$ [°C] | $T_{sink}$ [°C] | $\Delta T_{m,lift}$ [K] | COP [-] | Compressor Design | Absorber Design | Desorber Design | Ref. |
|---|---|---|---|---|---|---|---|---|---|---|
| **Wet compression cycle** | | | | | | | | | | |
| **Malewski** (Germany) | 1988 | 500 | 35 | 60 to 80 | n/a | 4.4 | wet, screw, s-st | Shell/tube horizontal | Shell/tube horizontal | [57] |
| **Bergmann, Hivessy** (Hungary) | 1990 | 1000 | 25 to 5 | 15 to 85 | 27.9 | 4.4 | wet, screw, s-st | Shell/tube horizontal | Shell/tube horizontal | [35] |
| **Torstensson, Nowacki** (Sweden) | 1991 | 1.4 | 16 to 3 | 35 to 60 | 38.6 | $3.0_C$ | wet, scroll, s-st | Tube/tube coaxial | Tube/tube coaxial | [58] |
| **Itard** (Netherlands) | 1998 | 13 | 44 to 38 | 40 to 53 | 5.3 | 3.1 | wet, liquid ring, s-st | Plate-fin vertical | Plate-fin vertical | [59] |
| **Zaytsev** (Netherlands) | 2003 | 18.9 | 70 to 65 | 76 to 92 | 16.3 | 1.4 | wet, twin screw, s-st | Shell/tube vertical | Shell/tube vertical | [60] |
| **ACHP with two-stage solution circuit** | | | | | | | | | | |
| **Rane, Radermacher** (USA) | 1991 | 4.2 | 4 to −5 | 96 to 104 | 100.5 | $1.0_C$ | dry, recip, t-st | Shell/tube vertical | Shell/tube vertical | [39] |
| **ACHP with single circuit and desorber/absorber heat exchange (DAHX)** | | | | | | | | | | |
| **Groll, Radermacher** (USA) | 1994 | 5 | 0 to −6 | 58 to 74 | 68.7 | $0.9_C$ | dry, recip, t-st | Shell/tube vertical | Shell/tube vertical | [42] |

**Explanations: Cap.:** Heating capacity; $\mathbf{\Delta T_{m,lift}}$: Total temperature lift determined as the difference of the logarithmic mean temperatures; **COP:** $_C$: Cooling COP; **Compressor design:** oil: Oil-lubricated; dry: Oil-free; wet: Lubrication done by solution; recip: Reciprocating compressor; s-st: Single-stage compression; t-st: Two-stage compression; **Absorber/Desorber design:** PHE: Plate heat exchanger; **Ref.:** Reference.

As initially described, the first commercially used ACHP systems were commissioned in recent years. Here, the ACHP cycle with single-stage solution circuit using a two-stage compression with intercooling and desuperheater before the mixing at the absorber inlet was implemented. The ACHP cycle with single-stage solution circuit is the most frequently experimental investigated cycle, accounting for 16 publications identified from 1983 until 2021. The installed capacities range from laboratory plants starting from 7 kW up to industrial pilot plants with capacities ranging from 160 kW up to 4500 kW. An identical inlet temperature was often selected for source and sink temperatures of laboratory systems, whereby a larger temperature glide was usually achieved on the sink side. The highest achieved sink temperature was at constant 115 °C during a steam generation process, with many other attempts frequently ranged below 100 °C due to the limitation of the open auxiliary circuits used. Due to large temperature glides occurring in the heat sink and source of ACHP cycles, the total temperature lift $\Delta T_{m,lift}$ is employed to evaluate the experimentally achieved temperature lifts, as suggested by Lorenz (1895) [61]. The $\Delta T_{m,lift}$ was determined as the difference of the logarithmic mean temperatures of the secondary fluids for heat sink and source using Equations (1) and (2):

$$\Delta T_{m,lift} = \Delta T_{m,log,Sink} - \Delta T_{m,log,Source} \tag{1}$$

$$\Delta T_{m,log} = \frac{T_{secondary\ fluid,in} - T_{secondary\ fluid,out}}{\ln \frac{T_{secondary\ fluid,in}}{T_{secondary\ fluid,out}}} \tag{2}$$

For the total temperature lift of the ACHP cycle with single-stage solution circuit, values ranging from −5.6 K to 38.7 K were achieved. As mentioned earlier, through the temperature glides occurring in the heat source and sink in combination with a large total temperature lift, a simultaneous use for cooling and heating demands is possible, as shown by Nordtvedt (2005) [44]. COPs of 2.1 to 11.3 were achieved, whereby a direct comparison is difficult due to the different temperature levels and stages of optimization (insulation, compressor size, pinch in heat exchanger). Regarding the compressor design, oil-lubricated reciprocating compressors with single or two-stage compression were used in most investigations. Two times twin screw compressors with oil lubrication and oil cooling were used. Only four cases of oil-free operation involving three reciprocating compressors (two single-stage and one two-stage compression) and one screw compressor were tested. The oil-lubricated two-stage reciprocating compressors often achieved the highest temperature lifts, proportional to the required pressure ratio. Here, intercooling with an additional heat exchanger between the compression stages was frequently used to reduce the occurring discharge temperature. In the absorber and desorber design, an evolution from shell-and-tube to corrugated plate heat exchangers (PHE) can be seen, and the alignment changed from horizontal to vertical. In this context, several innovative approaches have been employed. Mučić and Scheuermann (1984) [48] investigated a subdivision of the heat exchanger surface with linking solution pumps. The connection of several PHEs to extend the effective heat transfer length in combination with different operation modes, described as falling film and bubble mode, as well as mixing techniques were evaluated during the investigations by several authors [20,21,41,50].

The wet compression cycle has been investigated in five experimental studies. The first of these are industrial pilot plants with capacities of 500 kW and 1000 kW, which follow in time with the first installation of the single-stage ACHP pilot plants. Then a wet compression laboratory facility was investigated with one of the smallest capacities of all ACHP cycles at 1.4 kW. The tested temperature levels vary in all investigations with the shared feature of a larger temperature gradient over the heat sink. Single-stage compressors lubricated with the solution were used for all investigations, although a wide variety of different compressor types was tested. Except for the liquid ring compressor employed by Itard (1998) [59], all types were considered suitable for use in wet compression cycles. However, for efficient and reliable use, the authors highlight the importance of modifications, such as selecting suitable bearings and the design of the liquid injection

system [35,57,58,60]. For the design of absorber and desorber, an evolution from horizontal to vertical arrangement was again observed. Additionally, Torstensson and Nowacki (1991) [58] tested a coaxial arrangement of a tube-to-tube heat exchanger. Itard (1998) [59] employed plate-fin heat exchangers with the aim to increase the heat transfer area as a precursor to the commonly used corrugated PHE.

The ACHP with a two-stage solution circuit and the ACHP with a single-stage circuit with DAHX have been studied only once by a group of researchers led by Radermacher [39,42]. The researchers modified an existing single-stage laboratory facility designed by Rane et al. (1989) [51] to investigate the different cycle configurations as previously described. The ACHP with two-stage solution circuit investigated by Rane and Radermacher (1991) [39] had a capacity of 4.2 kW and achieved a total temperature lift of 100.5 K with almost identical temperature glides ranging from 4 °C to −5 °C for the source and from 96 °C to 104 °C for the sink. This enables simultaneous use for cooling and heating and results in a determined cooling COP of 1. Groll and Radermacher (1994) [42] achieved during the investigation of the ACHP with single-stage circuit with DAHX a capacity of 5 kW with a total temperature lift of 68.7 K with temperature glides ranging from 0 °C to −6 °C for the source and from 58 °C to 74 °C for the sink. Despite the lower temperature lift and required pressure ratio, the achieved cooling COP at 0.9 is smaller than the cooling COP obtained for the ACHP with two-stage solution circuit. For both investigations, oil-free two-stage reciprocating compressors with water cooling were used and vertically arranged shell-and-tube heat exchangers were installed as absorber and desorber. Groll and Radermacher (1994) [42] identified a potential COP increase of over 40% by optimizing the compressor for use in the ACHP with single-stage circuit with DAHX.

*Identified Existing Challenges*

Based on the experimental work presented and discussed in Table 1, the following challenges for the realization of ACHP systems can be identified:

- **Compressor discharge temperature:** Occurring discharge temperatures constrain the achievement of higher sink temperatures, as they can cause problems, such as the decomposition of used lubricating oils and material problems in the compressor.
- **Compressor lubrication:** When oil is used to lubricate the compressor, additional components are needed for separation and cooling, raising the complexity and costs. In addition, the oil tends to penetrate the whole circuit, which requires a recirculation system and can have a negative impact on the heat pump performance.
- **Oil-free operation of the system:** Compressors that can be operated oil-free or for which lubrication is done by the solution are often associated with higher equipment costs due to necessary modifications or are unavailable for commercial use.
- **Absorber and desorber design:** An efficient design of the absorber and desorber is an important factor in improving the performance of the system. Therefore, an advanced understanding of the occurring heat transfer phenomena is essential, especially for the absorber at high temperature and high-pressure operation.
- **Liquid–vapor mixing and distribution process:** When using PHE, an appropriate selection of the operation mode together with effective liquid–vapor mixing and distribution are important to achieve high overall heat transfer coefficients and system performances.
- **Solution pump:** Cavitation caused by changing low pressure conditions related to rapid changes in compressor operation and operating conditions is a major challenge for the solution pump in ACHP cycles due to the saturated liquid leaving the liquid–vapor separator.

## 4. Identification of Recent Developments and Possible Solutions

The current working domain of ammonia-water ACHP with single-stage solution circuit using components up to 28 bar and dry compression are between 0 °C (freezing of solvent) and 127 °C (calculation by Jensen et al. (2015) [10]). The upper limit originates in a

constrained compressor discharge temperature of 170 °C. The same calculations indicate that by using several high-pressure components, an extension of the upper limit to 193 °C is possible. Additionally, sink temperatures of more than 127 °C can be achieved with 28-bar components through measures such as oil cooling, liquid injection or wet compression. In the following, recent developments and possible solutions for the implementation of ACHP systems are identified based on the performed experimental work and accompanying theoretical studies. The grouping is based on the identified challenges and the order on the relevance to improve system performance and achievable parameters.

### 4.1. Compressor Solutions

To enable high temperature lifts and sink discharge temperatures, it is necessary to provide a high discharge pressure and pressure ratio. During compression of the ammonia vapor the discharge temperature is comparatively high due to the relatively low density and specific heat capacity of ammonia in the superheated vapor phase [62]. High discharge temperatures reduce the COP of the system and cause various problems, such as chemical decomposition of the working fluid, carbonization of the lubricant and collapse of seals [18]. Additionally, liquid portions can be included even for dry compression depending on the temperature and pressure level due to the characteristics of the ammonia-water mixture. Therefore, the compressor must be resistant to small amounts of liquid in the vapor stream during the compression. This is, furthermore, a fundamental requirement for wet compression. According to an extensive analysis by Zaytsev (2003) [60] and in agreement with the findings presented in Table 1, positive displacement compressors, such as reciprocating and screw compressors, have been identified as promising compressor solutions due to the higher achievable pressure ratios and the lower swept volumes when compared with dynamic compression systems. To reduce the superheat occurring during compression and maintain the discharge temperature in an acceptable range for the compressor, different potential compressor solutions are available for the implementation besides simple single-stage compression, as shown in Figure 4.

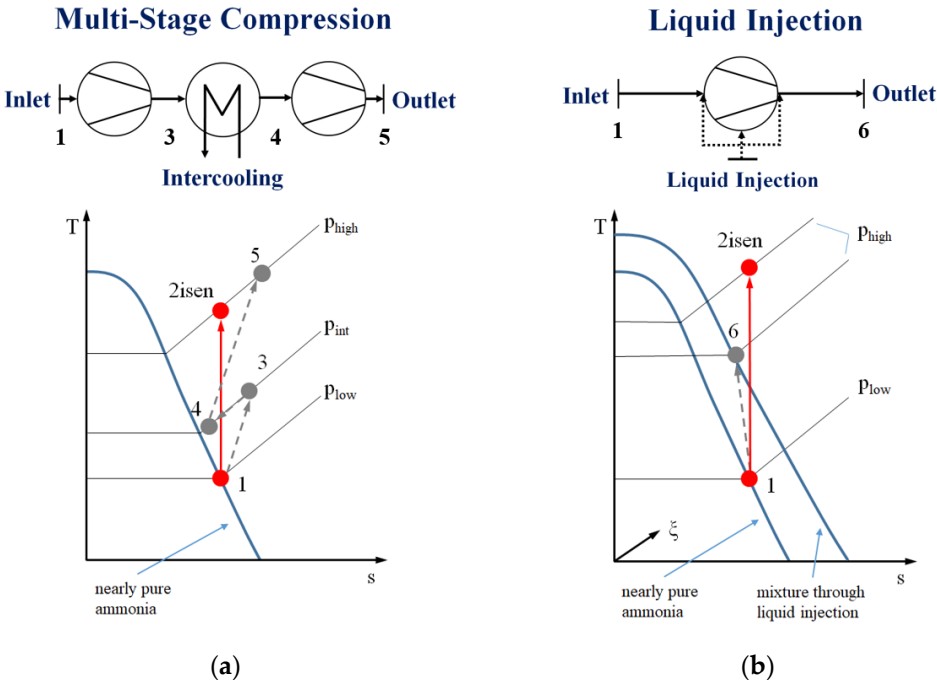

**Figure 4.** Different potential compressor solutions (simplified) illustrated with T-s diagrams of nearly pure ammonia including isentropic compression (solid red line) compared to: (**a**) Divided multi-stage compression with intercooling through an additional IHX and (**b**) single-stage compression with liquid injection (dashed grey lines).

The simplified compression processes are illustrated in T-s diagrams for nearly pure ammonia to demonstrate the proposed possibilities. The isentropic compression (from point 1 to $2_{isen}$) demonstrates the ideal compression process from the low-pressure to the high-pressure level. However, the achieved discharge temperature during real compression is significantly higher due to the occurring losses and irreversibility. Here, the identified maximum temperature for the compressor of 170 °C can be quickly exceeded. In the case of liquid injection, the composition of the working fluid changes during the compression process, adding another level to the representation.

As shown in Figure 4a, the first possible solution is to divide the compression into two or more stages with an additional IHX for cooling the vapor after each stage (from point 1 to 5). Therefore, multi-stage compression with intercooling can increase the achievable pressure ratio. The functionality of this solution with two-stage reciprocating compressors and intercooler connected to the lean solution circuit has been experimentally proven by various authors [22,44,45,54]. Moreover, there are other approaches for the implementation of intercooling. Jensen (2015) [18] investigated the effect of the positioning of the intercooler (upstream or downstream of the contained solution IHX between rich and lean solution) as well as other alternatives, such as the use of a bubble through intercooler or mixing option with liquid injection. He concluded that the solution with an additional IHX downstream the contained solution IHX can most effectively reduce the discharge temperature and costs for a multi-stage compression system.

Another promising solution, as shown in Figure 4b, is the use of a liquid-resistant single-stage compressor with the implementation of liquid injection (oil or oil-free using the working fluid) during the compression process (from point 1 to 6). The injection can take place at various compression grades (point of injection) and allows to provide different functions such as sealing, lubrication and lowering the vapor temperature during compression [63]. The point of injection and amount of the injected liquid is a matter of optimization [64]. If all the liquid is carried along with the vapor, it is defined as wet compression [34]. Bergmann and Hivessy (1990) [35] stated that the liquid injection makes the compressor operation smoother and more silent than dry compression. However, especially for operation as wet compression, the isentropic efficiencies obtained are often comparatively low due to the complexity of the compression process, causing various challenges in the design and optimization of the compressor and the injection system [65,66].

### 4.2. Absorber and Desorber Solutions

Absorber and desorber are critical components of ACHP cycles to achieve higher system efficiencies and obtain higher sink temperatures [54,55]. Many studies have been conducted to understand the characteristics of the occurring processes and thermodynamic properties [67,68]. For use in an ACHP system, various requirements are defined for the heat exchanger properties used. In addition to the general requirements, such as the reduction of size and pressure losses, the suitability for the desired operating parameters must be considered [52]. To reduce installation costs, compact heat exchangers with a high area density ($m^2/m^3$) of the heat transfer surface combined with a high overall heat transfer coefficient ($kW/(m^2K)$) are desired [69,70]. Furthermore, the establishment of effective liquid–vapor mixing and the complete and continual wetting of the heat transfer surfaces are important [71].

Based on the investigations carried out and the data presented in Table 1, the trend from horizontal to vertical shell-and-tube heat exchangers towards vertical PHE for use in ACHP systems can be recognized. The plates can be pressed with different shapes to increase turbulence, fluid distribution and surface area [72]. Due to these features, the design can be much more compact. In addition, PHE can provide high overall heat transfer coefficients, good wettability and liquid–vapor mixing [55]. Various approaches have been and are being investigated to improve the design and operation of absorber and desorber [73]. It was concluded that longer plates combined with good distribution of the mixture and operation in counter-flow with the coolant have positive effects on the

performance. Jung et al. (2014) [55] stated that increasing the ratio of plate length to gap (L/D) is more important than the ratio of width to gap (W/D) for more effective operation.

Several possibilities are available for the implementation. Due to the similarity of the operating conditions and requirements, the desorber is often designed and operated like a flooded evaporator. The absorber can be operated in falling film mode, where lean solution slides from top to bottom as a thin film on the surface of the vertical plate and vapor filling the free space is absorbed into the liquid film, releasing heat to the coolant. Another approach is to operate the absorber in bubble mode, where the lean solution and vapor are mixed at the bottom before entering the absorber and flow upwards in counter-flow with the coolant [74]. Furthermore, it is possible to operate several PHEs inline using different operating modes [24,54]. Figure 5 shows a schematic representation of a plate heat exchanger operated in counter-flow as absorber in falling film and bubble mode.

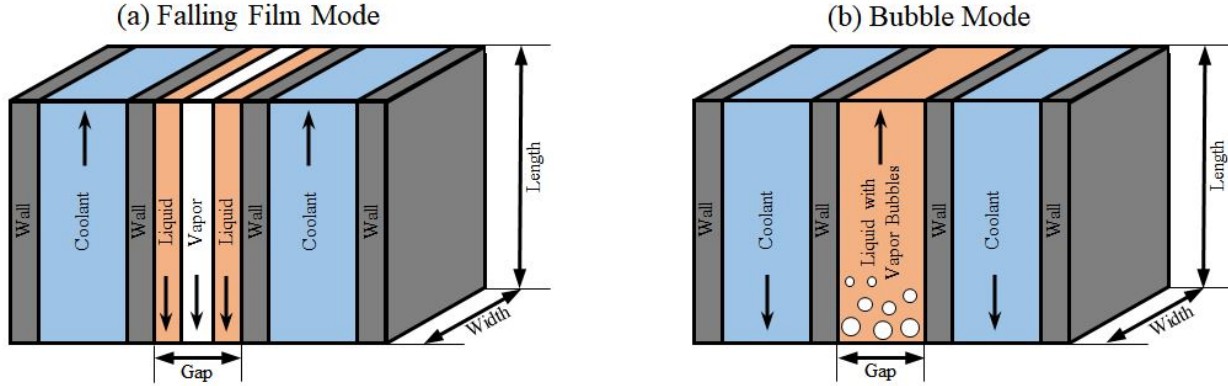

**Figure 5.** Schematic representation of a plate heat exchanger operated as absorber in (**a**) falling film and (**b**) bubble mode (adapted from Ahrens et al. (2019) [75]).

Bubble mode operation in the absorber of PHE in an ACHP cycle was investigated by An et al. (2013) [74]. They concluded that higher pressure is a more significant factor for increasing the heat exchanger performance than size based on experiments with different sized PHE. The maximum overall heat transfer coefficients achieved for different PHE sizes were between 0.96 and 1.61 kW/m$^2$K. In contrast, Lee et al. (2002) [76] reported values between 0.3 and 0.55 kW/(m$^2$K) at lower temperatures in a PHE test facility. More experimental data from Nordtvedt (2005) [77] for bubble and falling film absorber were between 0.6 and 1.4 kW/(m$^2$K) and for bubble desorber between 0.4 and 1.5 kW/(m$^2$K). Táboas et al. (2010) [78] stated for a bubble desorber, that pressure has only a slight influence and mass flux a major influence. Measured overall heat transfer coefficients varied between 2.5 and 4.1 kW/(m$^2$K). Due to many parameters affecting the absorption and desorption process, which are not all documented in these sources, a direct comparison or deducing trends becomes impossible. However, it can be concluded that the overall heat transfer coefficients can vary considerably depending on the given operating conditions and other factors such as the liquid–vapor distribution. Determining and predicting the required parameters more accurately for the design and controlling of ACHP cycles is an important goal to further disseminate ACHP technology for use in industrial applications.

### 4.3. Solution Pump Solutions

Cavitation is a major challenge for the solution pump in ACHP cycles. The saturated liquid coming from the liquid–vapor separator is very sensitive to changing low-pressure conditions caused by the compressor. The evaluation of the sources given in Table 1 shows that mainly centrifugal pumps for industrial pilot plants and diaphragm pumps for small laboratory facilities are used as solution pumps. To reduce the risk of cavitation, different approaches have been investigated. Besides the generally important reduction of pressure losses upstream of the pump inlet, the external subcooling of the lean solution, as used

by Rane et al. (1989) [51], is one possible solution. Additionally, experiments with an upstream booster pump or a special design of the separator to provide enough static height of the liquid level for the pressure increase upstream of the solution pump have been investigated by various researchers [44,55,56]. Furthermore, Risberg et al. (2004) [23] and Markmann et al. (2019) [56] introduced an option to control the liquid level in the high-pressure receiver by regulating the expansion valve to keep the low pressure stable and reduce the risk of cavitation due to rapid pressure changes.

### 4.4. Alternative Working Fluid Pairs

In general, all working fluid pairs suggested for absorption heat pumps may also be used in ACHP cycles [79]. The unique feature of a working fluid pair is a greater temperature glide during complete evaporation and condensation compared to the glide of heat source and sink, causing the partial phase change that makes a solution circuit necessary. For a comprehensive review and possible identification of interesting alternatives, Table 2 lists an overview of literature sources investigating alternative working fluid pairs besides ammonia-water for use in ACHPs. There are mainly theoretical studies with few experimental investigations on a laboratory scale (below 30 kW heating capacity).

Following the Montreal Protocol and Kyoto Protocol addressing the ozone depletion potential (ODP), the main issues in recent years have been the EU Regulation No. 517/2014 (2014) [80], also known as F-Gas Regulation, and the European Directive 2006/40/EC (2006) [81], establishing very strict limits on the GWP values of applicable refrigerants. Due to this legislation, many sources dealing with substances having an ODP > 0 or a GWP > 1000 are not listed and will not be discussed further. The Globally Harmonized System of Classification and Labelling of Chemicals (GHS) classification was carried out using the "GESTIS-Stoffdatenbank" [82]. Maximum workplace concentration (MAK) values are taken from the "DFG MAK- und BAT-Werte-Liste 2020" [83]. The applicable temperature range is defined so that saturation pressure is between 1 and 25 bar. If 200 °C is still possible at the sink with a vapor pressure below 25 bar, 200 °C is assumed to be the upper limit.

**Table 2.** Overview of alternative working fluid pairs proposed in the literature for use in ACHP cycles.

| Author (Location) | Year | Exp. | Refrigerant | GHS | Absorbent | GHS | $T_{min}/T_{max}$ | Ref. |
|---|---|---|---|---|---|---|---|---|
| **Inorganic Refrigerant** | | | | | | | | |
| **Åhlby, Hodgett** (Sweden) | 1990 | | $NH_3$ [1] | ⬦⬦⬦⬦ | $H_2O$ [2]/LiBr [3] | -/⬦ | 0/200 | [84] |
| **Chatzidakis, Rogdakis** (Germany) | 1992 | | $NH_3$ [1] | ⬦⬦⬦⬦ | $H_2O$ [2]/LiBr [3] | -/⬦ | 0/200 | [85] |
| **Tarique, Siddiqui** (India) | 1999 | | $NH_3$ [1] | ⬦⬦⬦⬦ | NaSCN [4] | ⬦⬦ | −35/160 | [86] |
| **Hannl** (Austria) | 2015 | X | $NH_3$ [1] | ⬦⬦⬦⬦ | $LiNO_3$ [5] | ⬦⬦ | −40/200 | [87] |
| **Herold et al.** (USA) | 1991 | | $H_2O$ [2] | - | LiBr [3] | ⬦ | 0/200 | [88] |
| **Ansari et al.** (India) | 2018 | | $H_2O$ [2] | - | LiBr [3] | ⬦ | 0/200 | [89] |
| **Gudjonsdottir et al.** (Netherlands) | 2017 | | $NH_3$ [1]/$CO_2$ [6] | ⬦⬦⬦⬦ / ⬦ | $H_2O$ [2] | - | 0/200 | [90] |
| **Groll, Kruse** (Germany) | 1992 | X | $CO_2$ [6] | ⬦ | Acetone [7] | ⬦⬦ | −40/80 | [91] |
| **Moreira-da-Silva et al.** (Spain) | 2019 | | $CO_2$ [6] | ⬦ | Acetone [7] | ⬦⬦ | −40/80 | [92] |
| **Aldás** (Spain) | 2020 | | $CO_2$ [6] | ⬦ | Acetone [7] | ⬦⬦ | −40/80 | [93] |

**Table 2.** *Cont.*

| | | | Organic Refrigerant | | | | | |
|---|---|---|---|---|---|---|---|---|
| **Endo et al.** (Japan) | 2007 | | DME [8] | ⬥⬥ | MeOH [9] | ⬥⬥⬥ | −30/200 | [94] |
| **Kawada et al.** (Japan) | 1991 | X | TFE [10] | ⬥⬥⬥⬥ | TEGDME [11] | ⬥ | 75/200 | [95] |
| **Nogues et al.** (Spain) | 1997 | | TFE [10], MeOH [9] | ⬥⬥⬥⬥ , ⬥⬥⬥ | TEGDME [11] | ⬥ | 65/200 | [96] |
| **Bourouis et al.** (Spain) | 2000 | | TFE [6]/H$_2$O [2] | ⬥⬥⬥⬥ / - | TEGDME [11] | ⬥ | 75/200 | [97] |
| **Mestra et al.** (Spain) | 2003 | X | TFE [10], MeOH [9], | ⬥⬥⬥⬥ , ⬥⬥⬥ , | PEGDME 500 [12] | n/s | 65/200 | [98] |
| | 2005 | | HFIP [13] | ⬥⬥ | | | | [99] |

**Explanations: Exp.:** Experimental work available; **GHS:** Globally Harmonized System of Classification and Labelling of Chemicals; **T$_{min}$/T$_{max}$:** Applicable temperature range; n/s: not specified; CAS: Chemical Abstracts Service; MAK: Maximum workplace concentration; [1] **Ammonia (NH$_3$)** | 7664-41-7—CAS Registry Number | 14 (mg/m$^3$)—MAK value; [2] **Water (H$_2$O)** | n/s—CAS Registry Number | n/s (mg/m$^3$)— MAK value; [3] **Lithium bromid (LiBr)** | 7550-35-8—CAS Registry Number | n/s (mg/m$^3$)—MAK value; [4] **Sodium thiocyanate (NaSCN)** | 540-72-7—CAS Registry Number | n/s (mg/m$^3$)—MAK value; [5] **Lithium nitrate (LiNO$_3$)** | 7790-69-4—CAS Registry Number | n/s (mg/m$^3$)—MAK value; [6] **Carbon dioxide (CO$_2$)** | 124-38-9—CAS Registry Number | 9100 (mg/m$^3$)—MAK value; [7] **Acetone** | 67-64-1—CAS Registry Number | 1200 (mg/m$^3$)—MAK value; [8] **Dimethyl ether (DME)** | 115-10-6—CAS Registry Number | 1900 (mg/m$^3$)—MAK value; [9] **Methanol (MeOH)** | 67-56-1—CAS Registry Number | 130 (mg/m$^3$)— MAK value; [10] **2,2,2-Trifluoroethanol (TFE)** | 75-89-8—CAS Registry Number | n/s (mg/m$^3$)—MAK value; [11] **Tetraethylene glycol dimethyl ether (TEGDME)** | 143-24-8—CAS Registry Number | n/s (mg/m$^3$)—MAK value; [12] **Polyethylene glycol dimethyl ether (PEGDME 500)** | 24991-55-7—CAS Registry Number | n/s (mg/m$^3$)—MAK value; [13] **Hexafluoroisopropanol (HFIP)** | 920-66-1—CAS Registry Number | n/s (mg/m$^3$)—MAK value.

There were 13 alternative working fluid pairs investigated in the literature, divided into inorganic and organic refrigerants with TFE and HFIP as the only non-natural refrigerants. Except for $CO_2$/Acetone and $NH_3$/NaSCN, the applicable temperature range is always 200 °C or higher. However, the minimum temperature depends strongly on the refrigerant selected and is generally lower for inorganic refrigerants. A further comparison of the cycle performance among the working fluid pairs is not appropriate due to the large differences in vapor pressure.

Åhlby and Hodgett (1990) [84] as well as Chatzidakis and Rogdakis (1992) [85], added LiBr as absorbent to the ammonia-water mixture and studied the suitability of the three-component mixture for the ACHP cycle. They stated inconsistent results regarding the improvement potential and recommended further investigations. Tarique and Siddiqui (1999) [86] and Hannl (2015) [87] investigated $NH_3$ paired with salts to increase the amount of refrigerant in the gas phase, eliminating the separation that may be required at increased source temperatures to reduce the water content in the vapor phase. Conducted experiments demonstrated the general functionality of this approach, although further research is required to establish prototypes and applications. The investigations on $H_2O$/LiBr in a temperature range between 2 °C and 35 °C, done by Ansari et al. (2018) [89], and in double-effect cycles, done by Herold et al. (1991) [88], are both far from practical application. Gudjonsdottir et al. (2017) [90] paired $NH_3$ with $CO_2$ and $H_2O$ in a wet compression cycle. The simulations of these cycles predict advantages in efficiency by increasing the amount of $CO_2$. However, there are many uncertainties like the formation of solid phases, ternary property data and the compressor model used so that the authors suggest experimental validation as a necessary next step. $CO_2$/Acetone is an interesting working fluid pair for safety aspects, but with standard components only feasible below 80 °C. Experiments by Groll and Kruse (1992) [91] and Groll (1994) [100] revealed poor heat transfer properties with the coaxial heat exchangers used and little further experience with this working fluid pair in the literature. In recent years, the thermodynamic properties have been increasingly studied experimentally, and Moreira-da-Silva et al. (2019) [92] simulated the usability for ACHP cycles in automotive air conditioning based on the available composition data. Aldás (2020) [93] conducted thermodynamic modelling and simulation of the cycle and theoretical-experimental investigations of the desorption process in a PHE.

DME/MeOH was theoretically investigated by Endo et al. (2007) [94] in the temperature range between 7 °C and 60 °C for the single stage cycle and the DAHX cycle. The differences in efficiency between pure DME and the mixture in both cycles were small. They suggest to further proof their findings in experiments. TEGDME or PEGDME as solvent use flammable and toxic TFE or MeOH as the refrigerant. These mixtures are suitable for temperatures above 65 °C and PEGDME for cases where the compressor may be lubricated by the lean solution [98]. Kawada et al. (1991) [95] stated that TFE/TEGDME shows good thermal stability and no corrosion on copper or carbon steel up to 180 °C. Nogues et al. (1997) [96] investigated the performance for TFE/TEGDME and MeOH/TEGDME for use in the single-stage and DAHX cycles. They reported interesting features for use in high temperature applications with better results for TFE/TEGDME and suggested further investigations. Bourouis et al. (2000) [97] conducted further investigations for a three-component mixture with additional water and reported further benefits such as an improvement in performance. Mestra et al. (2003) [98] investigated the working pairs TFE/PEGDME and MeOH/PEGDME to develop a 15 kW pilot plant. Here, MeOH was selected as the refrigerant for further investigations due to the lower pressure ratio required. In addition, Mestra (2005) [99] investigated the working pair HFIP/PEGDME, which was not tested experimentally. Besides theoretical studies, so far, there is only little experimental experience with the use of alternative working fluid pairs in ACHP cycles. Therefore, further investigations are essential for a successful implementation.

### 4.5. Summary of Existing Solutions

Figure 6 summarizes recent developments and existing solutions for the realization of ACHP systems. The classification is based on the previously presented and discussed solutions. The fields highlighted in green indicate the currently most frequently used solutions for commercial applications of each system part. All other solutions have at least been tested on a laboratory scale and often need further investigation for commercial use.

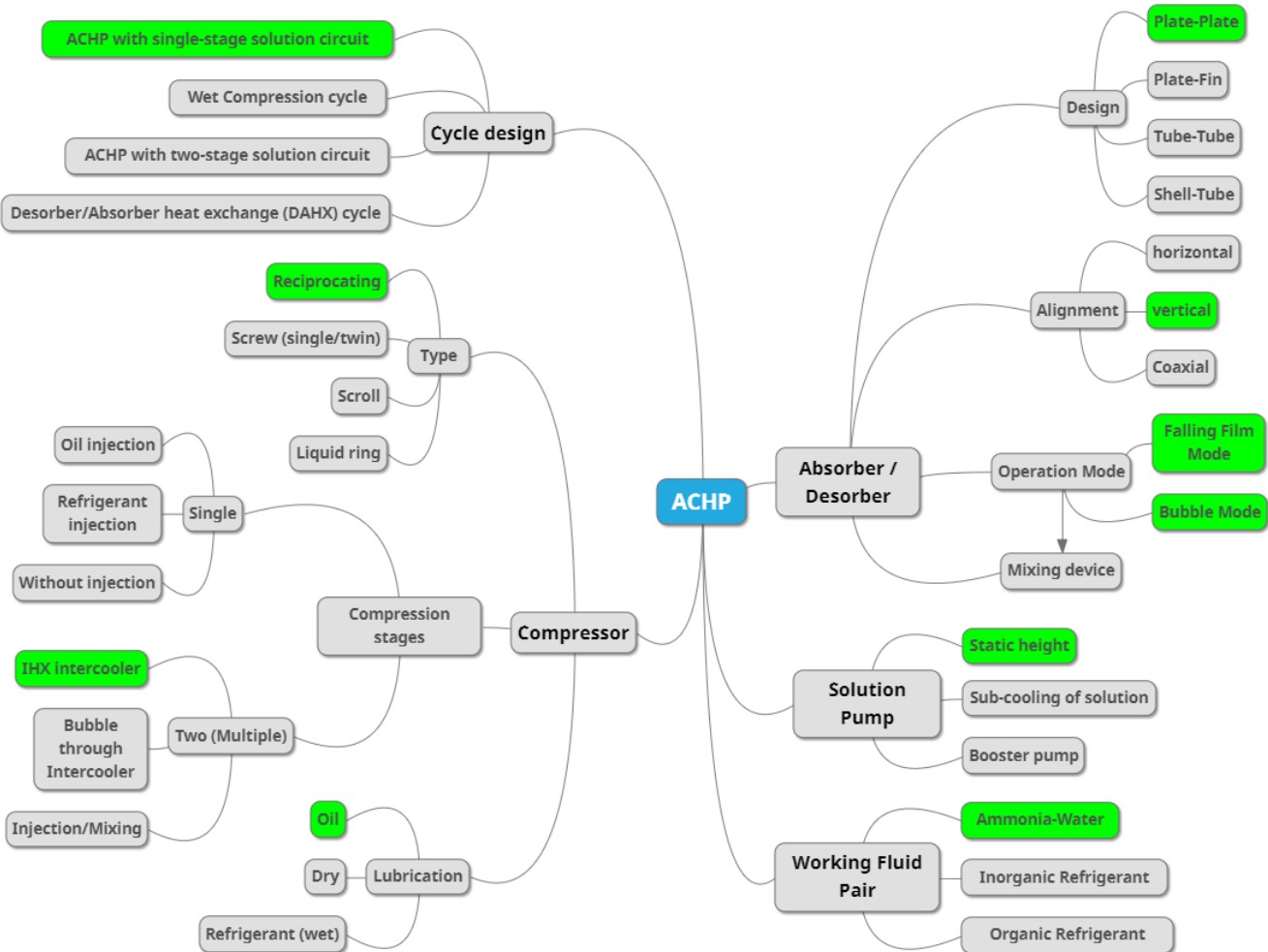

**Figure 6.** Overview of recent developments and existing solutions for the realization of ACHP systems (fields highlighted in green indicate the currently most frequently used solutions for commercial applications of each system part).

Four different cycle configurations were discussed for the design of the ACHP system, whereby the ACHP with single-stage solution circuit was most frequently investigated and is used commercially today. The compressor, as a major component and constraint for achieving higher sink discharge temperatures, features a wide range of solutions. Against the background of reducing occurring compressor outlet temperatures and costs, several types with different compression stages and lubrication solutions were used and studied. For commercial applications, oil-lubricated reciprocating compressors with two-stage compression and intercooling are currently employed. For the absorber and desorber, vertically aligned PHEs are used in different operating modes with the associated mixing devices. The specific design and operation remain a subject of ongoing research to further increase the efficiency while reducing the required size and costs. In addition to the general low placement of the solution pump and the design of the pipes with as little pressure loss as possible, a high static height of the liquid level upstream the pump inlet is achieved by the strategic design of the liquid–vapor separator. There have been many primarily

theoretical studies on alternative working fluid pairs, yet experimental experience for use in ACHP systems is limited to ammonia-water. The use and combination of other presented solutions are explicitly not excluded to overcome existing challenges. Depending on the circumstances, a follow-up and combination of other solutions can be a promising approach to improve the achievable operating parameter and efficiency to increase the suitable application range and competitiveness compared to VCHPs and conventional solutions for industrial high temperature applications.

## 5. Future Trends

In recent years, there has been a growing demand in the utilization of ACHPs with ammonia-water as working fluid for high temperature industrial applications [18,25]. The research focuses on achieving higher sink temperatures and system efficiencies to increase the suitable application range and competitiveness compared to VCHPs and conventional solutions. Here, the aim is to describe possible future trends to overcome the identified challenges by using conducted research and recent developments and solutions as a source of ideas for potential approaches for further research. Various approaches are used for achieving this aim:

- For the general system design, the ACHP with single-stage solution circuit has been used commercially. Nevertheless, there is still potential for further improvements, such as the optimized system control in case of varying operating conditions [18].
- To increase the application range and competitiveness of the ACHP, cost-effective and simple compressor solutions will be utilized. Besides using multistage oil-lubricated reciprocating compressors, the single-stage screw compressors with liquid injection and lubrication with the solution will be implemented as an alternative. In contrast to dry compression, the advantages of wet compression with reduced superheating and low discharge temperature can be beneficial [63]. In addition, the removal of the lubricating oil can lead to a reduction in the complexity and cost of the system. However, recent studies by Ahrens et al. (2019) [75,101] and Gudjonsdottir et al. (2019) [66] have indicated that suitable compressors are not yet commercially available and require further research and development. Insights and findings can be gained from related studies dealing with oil (Wu et al. (2017) [102]) or pure refrigerants as ammonia (Tian et al. (2017) [65]) and water (Wu et al. (2020) [5]).
- For the design and operation of absorber and desorber, there is a clear trend towards vertical PHE. This is often associated with factors such as compact design and cost-effective production. However, important factors for the efficiency, such as the liquid–vapor mixing and distribution, are often difficult to determine and thus challenging to predict for varying operating parameters [73]. The application of additive manufacturing techniques to produce plates and the liquid–vapor distribution may offer interesting possibilities. Solving the distribution challenge and being able to determine and predict the required parameters more accurately for the design and controlling of ACHP cycles, especially considering the desired high temperature operation, is an important goal of further research.
- In addition to the static height, a possible sub-cooling of the lean solution upstream of the inlet can result in a sustainable operation of the solution pump.
- For the use of alternative working fluid pairs in ACHP cycles, there is so far only little experimental experience apart from theoretical studies besides ammonia-water, as presented in Section 4.4. However, the use of alternative working fluid pairs is a promising solution.

In summary, different research trends for the use of ACHP systems at high temperature operation can be identified. To address the identified challenges and increase the application range and competitiveness with respect to other systems, future research should focus on the development of (oil-free) liquid-injected compressors and the efficient design and operation of the absorber and desorber.

## 6. Conclusions

This work presents the current state of technology and aims to identify existing challenges and future trends for the utilization of ACHPs at high temperature operation. Different modifications of the ACHP cycle were presented and discussed in detail. Furthermore, a comprehensive overview of conducted experimental work was given concerning the described cycle modifications and existing challenges were identified. Recent developments and possible solutions were discussed based on current research activities and summarized in a detailed mind map. Finally, possible future trends for further research activities were defined. The following conclusions for the ACHP technology can be drawn:

- Interest in the ACHP system has grown in recent years and has been supported by the successful implementation of first commercial units.
- Many studies, both theoretical and experimental, focused on improving the achievable operating parameter and system efficiency to increase the competitiveness compared to VCHPs and conventional solutions for industrial high temperature applications.
- Various configurations of the ACHP system with special characteristics have been developed and studied. The ACHP with a single-stage solution circuit was most widely investigated and is the only configuration in commercial use today. With the aim of further optimization of the ACHP system, existing challenges for the main components such as compressor, absorber/desorber and solution pump were identified based on the conducted research.
- Compressors used so far have been positive displacement compressors such as reciprocating, screw and scroll. For the achievable parameters of the ACHP system, the compressor is a constraining component and associated with challenges, such as discharge temperature, lubrication, efficient and oil-free operation of the system. A variety of possible solutions to address these challenges, such as multi-stage compression with intercooling or liquid injection, were investigated.
- Different types and configurations of heat exchangers have been used. For the design and operation of absorber and desorber, a clear trend towards vertically arranged PHEs has emerged. However, there are still challenges associated with the optimal design to achieve good liquid–vapor distribution and achieving high overall heat transfer coefficients. The ability to determine and predict the parameters required for the design and control of absorber and desorber more accurately is an important goal in further disseminating ACHP technology for use in industrial applications.
- Various strategies to avoid cavitation in the solution pump have been successfully implemented and tested, thus this problem discussed in earlier literature was solved.
- In addition to ammonia-water, a variety of alternative working fluid pairs for use in ACHP systems have been investigated. Further research and development are required to evaluate the reported potential improvements in a practical application.
- Based on the conducted investigations as well as recent developments and solutions, the future trends for further research were defined for all identified challenges.
- For the increased use of ACHP systems in high temperature applications, future research should focus on the development of (oil-free) liquid-injected compressors and the efficient design and operation of the absorber and desorber.

**Author Contributions:** Conceptualization, M.U.A., M.L., I.T., A.H., S.K., R.W. and T.M.E.; methodology, M.U.A., M.L. and I.T.; investigation, M.U.A., M.L. and I.T.; resources, M.L., A.H., S.K., R.W. and T.M.E.; writing—original draft preparation, M.U.A.; writing—review and editing, M.U.A., M.L., I.T., A.H., S.K., R.W. and T.M.E.; visualization, M.U.A.; supervision, I.T., A.H., S.K., R.W. and T.M.E.; project administration, M.U.A.; funding acquisition, A.H., R.W. and T.M.E. All authors have read and agreed to the published version of the manuscript.

**Funding:** This research was funded by Research Council of Norway as part of HighEFF—Centre for an Energy Efficient and Competitive Industry for the Future, an 8-year Research Centre under the FME-scheme (Centre for Environment-friendly Energy Research, 257632).

**Institutional Review Board Statement:** Not applicable.

**Informed Consent Statement:** Not applicable.

**Data Availability Statement:** Data sharing not applicable.

**Acknowledgments:** The work is part of HighEFF—Centre for an Energy Efficient and Competitive Industry for the Future, an 8-year Research Centre under the FME-scheme (Centre for Environment-friendly Energy Research, 257632). The authors gratefully acknowledge the financial support from the Research Council of Norway and user partners of HighEFF.

**Conflicts of Interest:** The authors declare no conflict of interest.

## Abbreviations

The following abbreviations are used in this manuscript:

| | |
|---|---|
| ACHP | Absorption–compression heat pump |
| CAS | Chemical Abstracts Service |
| CFC | Chlorofluorocarbons |
| COP | Coefficient of performance |
| DAHX | vapor compression cycle with solution circuit and desorber/absorber heat exchange |
| GHG | Greenhouse gas |
| GHS | Globally Harmonized System of Classification and Labelling of Chemicals |
| GWP | Global warming potential |
| HTHP | High temperature heat pump |
| IHX | Internal heat exchanger |
| MAK | Maximum workplace concentration |
| ODP | Ozone depletion potential |
| PHE | Plate heat exchanger |
| VCHP | Vapor compression heat pump |

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
