# Peer review of "Identification of Existing Challenges and Future Trends for the Utilization of Ammonia-Water Absorption–Compression Heat Pumps at High Temperature Operation"

_applsci, doi:10.3390/app11104635_

Round 1

Reviewer 1 Report

The paper presents reviews of absorption-compression air-conditioning system using a zeotropic mixture of water and ammonia as a working fluid.  The paper is well written, however some revisions are required for publication. 

  1. It is better to avoid general things in the abstract and describe important findings and key content obtained through the literature reviews.
  2. In Fig. 4, add thermodynamic cycle diagrams for each solution.
  3. Fig. 5 showed only counter flow type conceptual plate heat exchangers, but it will be better to show typical samples (cases) of actual applied heat exchangers.  

Reviewer 2 Report

Applsci-1203555

Comments on “Identification of Existing Challenges and Future Trends for the Utilization of Ammonia-Water Absorption-Compression Heat Pumps at High Temperature Operation”

  • Check the English and remove minor typos, and grammatical errors from the paper.
  • There are other processes recently developed on ammonia production at high temperature conditions. Authors should review state-of-the-art technologies on clean fuel production such as synthetic fuel or ammonia production and highlight the advantages of the system they propose in the paper. For example, searching the literature, following paper are suggested to be read and used: Comparative study of the performance of air and geothermal sources of heat pumps cycle operating with various refrigerants and vapor injection. Investigation of the effect of using various HFC refrigerants in geothermal heat pump with residential heating applications.  Determination of the optimal discharge pressure of the transcritical CO2 heat pump cycles for heating and cooling performances based on new correlation. 
  • Table 2: Can authors elaborate more information on systems compared in this table? Such as energy capacity (e. g. MWth, size, production level, CO2 emission).
  • Can authors add technology readiness level to the systems given in Table 6.
  • Any suggestions for future research in the area?
  • There are some abbreviations that have been used without introducing it in the paper at first place.

All in all, the paper can be accepted for publication once above comments are addressed.

Round 2

Reviewer 2 Report

  • Please recheck all references & the paper. For example, “2” in “CO2” should be subscript.
  • Why are some references written 100% in capital format?
  • The content of Table 2 is not clear.
  • Please confirm authors have drawn all figures in the manuscript. Otherwise, please obtain the required permission for figures.
